# Non-adherence to the World Health Organization's physical activity recommendations and associated factors among healthy adults in urban centers of Southwest Ethiopia

**Sabit Zenu**[1]*, **Endegena Abebe**[2], **Mohammed Reshad**[1], **Yohannes Dessie**[3], **Rukiya Debalke**[1], **Tsegaye Berkessa**[1]

**1** Department of Public Health, College of Health Sciences, Mettu University, Mettu, Ethiopia, **2** Department of Biomedical Sciences, College of Health Sciences, Mettu University, Mettu, Ethiopia, **3** Department of Nursing, College of Health Sciences, Mettu University, Mettu, Ethiopia

* sabitzeinu91@gmail.com

**Data Availability Statement:** The necessary data are within the Supporting Information file.

## Abstract

Physical inactivity is a major risk-factor of non-communicable diseases. The World Health Organization has set physical activity recommendations for adults to reduce physical inactivity and its consequences. However, 1.4 billion adults are non-adherent to the recommendation worldwide. The prevalence of non-adherence to this recommendation and its predictors has not been assessed in urban Ethiopia. This study aimed to determine the prevalence of non-adherence to physical activity recommendations and identify its associated factors among healthy adults in urban centers of Southwest Ethiopia. **A community**-based cross-sectional study was employed from May to June 2021, involving 1191 adults in urban centers of Southwest Ethiopia. Data was collected using Global Physical Activity Questionnaire. Multivariable logistic regression was used to identify factors associated with non-adherence to physical activity recommendations using 95% confidence interval of adjusted odds ratio at P-value of < 0.05.Overall, 61.2% of participants were non-adherent to physical activity recommendations. Older age (AOR = 6.6; 95%CI (2.3–19)), female sex (AOR = 6.1; 95%CI (3.5–10.5)), lower educational status (AOR = 0.5; 95%CI (.28–0.93)), less community engagement (AOR = 2.7;95% CI (1.3–5.5)), lower level of happiness (AOR = 4.7; 95% CI (1.3–16.8)) and physical inactivity of family members (AOR = 2.5; 95%CI (1.4–4.3)) were associated with non-adherence. The prevalence of non-adherence to physical activity recommendations in the study area is high. Age, sex, educational status, community engagement, level of happiness and physical inactivity of family members were predictors of non-adherence to the recommendations. Interventions have to target females and older adults. Community participation and family based physical activity have to be advocated to avert the consequences of physical inactivity.

**Funding:** The authors received no specific funding for this work.

**Competing interests:** The authors have declared that no competing interests exist.

## Introduction

Physical inactivity (PI) is a major behavioral risk factor of non-communicable diseases (NCDs), and it is the fourth leading cause of death worldwide. PI is raising around the world posing serious threat for people's health by contributing to the increasing prevalence of NCDs. PI is thought to be the primary cause of 21–25 percent of breast and colon cancer, 27 percent of diabetes, and 30 percent of ischemic heart disease [1, 2]. Regular physical activity (PA) helps to minimize the risk of NCDs [1, 3]. In addition, it reduces symptoms of depression and anxiety; enhances thinking, learning, and judgment skills. Furthermore, PA improves energy balance, and aids in weight management, which helps to combat the global NCD pandemic [2].

The World Health Organization (WHO) has issued global physical activity recommendation (PAR) in 2010 and updated it with further emphasis in 2020. The organization recommends healthy adults to perform at least 150 minutes of moderate, or 75 minutes of vigorous intensity aerobic physical activity throughout a week; or to do an equivalent combination of moderate and vigorous activities. The recommendation can also be met if adults accumulate at least 600 metabolic equivalents in a week. The organization has called on member states to put special emphasis on creating suitable and healthy transportation, and suitable environments that promote physical activity. In 2018, WHO launched a new Global Action Plan on Physical Activity which outlined four policy action areas, 20 specific policy recommendations and actions for various stakeholders, to increase PA worldwide [3–5]. The organization has also introduced ACTIVE toolkit in 2019 that provides specific technical guidance on how to start and implement the 20 policy recommendations outlined in the global action plan [6].

The PAR and global action plan are meant to promote health, individual wellbeing and prevention of NCDs. PI, or performing less than the recommended level of PA leads to several unwanted health consequences including the increased risk of NCDs [1, 2, 7, 8]. Despite this recommendation and initiatives, 1.4 billion adults are doing less than the recommended levels of PA. Worldwide, one in three women and one in four men do not do enough PA to stay healthy. Levels of PA are reported to be twice as high in high-income countries compared to low-income countries. Furthermore, the global levels of PA didn't show any improvement over the last two decades. Contrarily, insufficient activity increased by 5% (from 31.6% to 36.8%) in high-income countries between 2001 and 2016 [3, 5].

The level of non-adherence to PAR among adults is also high in developing countries. In Malaysia, more than four in ten of adult population is categorized as non-adherent to PAR [9]. The magnitude stands at 34% in Papua New Guinea [10] and much higher in Kerala, India where the level of non-adherence to PAR is as high as 90.3% [11]. In African countries, three quarters of adult population are doing less than recommended level of PA. In Nigeria, the prevalence is as high as 77.8% among university employees [12] and 69.7% among civil servants [13]. The Kenyan national STEPS survey included 4066 study participants in 2015; among these, 80.3% are categorized as physically inactive and are not meeting current PAR [14].

In Ethiopia, most studies assessed the problem of non-adherence to health guidelines, including PAR, among the already diseased populations. These studies assessed the level of non-adherence to PAR mainly among hypertensive and diabetic patients [15–17]. National STEPS survey in 2015 reported the level of non-adherence to PAR to be as low as 6%. In addition, a study conducted to assess the level of non-adherence to PAR among public servants in Northern Ethiopia reported prevalence of the problem to be 41% [18, 19]. Non-adherence to PAR has negative impacts on individuals, health systems, economic development, community well-being and quality of life. Despite this, little is known on the prevalence of non-adherence to the current recommendations in urban centers of Ethiopia. In

addition, advent of COVID-19 pandemic has resulted in a unique challenge to health of population including physical exercise. This study aimed to determine level of non-adherence to WHO's physical activity recommendations and identify its associated factors among adult urban residents in Southwest Ethiopia.

## Materials and methods

### Study area and period

The study was conducted among adults in three major towns of Southwest Ethiopia; Bedelle, Mettu and Gambella. Bedelle and Mettu towns are capitals of Ilu Aba Bor and Buno Bedelle Zones in Oromia region respectively. Gambella town is the regional capital of Gambella region.

Community based cross sectional study was conducted from May to June, 2021.

### Sample size determination and sampling technique

Sample size was calculated for both objectives by using STATCAL of Epi Info-7, and the larger sample size was used. The larger sample size was obtained\ by using the prevalence of non-adherence to PAR among adults in Northern Ethiopia as 41%, 95% CI, margin of error of 0.04, design effect of two and adding 10% for non-response. This calculation resulted in sample size of 1277. Multi-stage sampling was used to reach on respondents. From the three towns, lowest administrative units (Kebeles) were selected by lottery method. After developing list of households in selected kebeles, samples were proportionally allocated to each kebeles and systematic random sampling technique was employed to reach on households depending on house numbers. From selected households, one individual was selected from eligible adults by Kish method.

### Data collection tools and procedures

PA was measured using the second version of Global Physical Activity Questionnaire (GPAQ). The participants were asked about their physical activities by using GPAQ translated to the local language, Afaan Oromo, with the help of language experts. Information on socio demographic characteristics of participants, family practices and social characteristics were collected using questionnaires from the EDHS 2015 [20]. Social capital and related variables were measured by using the integrated social capital measurement tool validated for use in developing countries [21]. Trained nurses collected the data. Show-cards were used as per the recommendation of GPAQ to indicate examples of activities categorized in the three dimensions of PA.

### Data analysis procedures

After collection, data were entered to epidata version 3.1 and exported to SPSS version 20 for analysis. The GPAQ questionnaire assessed PA in three dimensions. These dimensions are: work related PA, recreational PA and travel or transportation related PA. All dimensions were converted in to the Metabolic Equivalents. Individuals who achieved less than 600 Metabolic Equivalents in a week were categorized as non-adherers to the WHO recommendation of PA. Proportions and other descriptive statistical analysis were used to describe the data. Multivariable logistic regression was used to identify factors associated with non-adherence to PAR. Associations between non adherence and independent variables were declared by using the 95% CI for the AOR at $p$-value of 0.05.

### Ethics statement

This research was conducted according to the principles of Declaration of Helsinki. The proposal and conduct of the study was ethically cleared by Mettu University ethics review

committee. Written informed consent was taken from selected participants. All information provided by the participants was kept confidential. In addition, any information leading to identification of study participants was not included in data collection tool.

## Result

### Socio demographic characteristics of participants

Among 1277 total sample size, 1191 adults volunteered to participate in the study resulting in 93.3% response rate. Most of the participants were in the age group of 30–44 years. Females accounted for more than half of the participants 644(54.0%), while more than seven in ten of participants were ethnic Oromo 847(71.1%). Orthodox Christianity 433(37.8%), Protestant 373(31.7%) and Islam 347(28.7%) were the predominant religions of the participants (Table 1).

### Non-adherence to the WHO recommendations of physical activity

From the overall study participants, 729(61.2%), 95% CI (56.2–66.0) failed to meet the WHO recommendation for physical activity. The level of non-adherence to PAR is higher among females when compared with males (*chi square tests* = 37.4; df = 1, $p$ <0.001). Non-adherence to PAR steadily increases as age increases from 46.2% among age group 18–29 to 85% among age group of 70 years and above(*chi square test* = 14.4; df = 4, $p$ = 0.006).

### Factors associated with non-adherence to physical activity recommendations

On the final multivariable model; older age, female sex, lower educational status, not having family members who do physical exercise, not participating in community activities and poor self-reported level of happiness is associated with non-adherence to PAR (Table 2).

In this study, old aged participants are nearly seven times more likely to be non-adherent to PAR when compared with younger participants; AOR = 6.6: 95% CI [2.3–19]. In addition, female participants are six times as non-adherent as their male counterparts; AOR = 6.1:95%

**Table 1. Socio demographic characteristics of participants in urban centers of Southwest Ethiopia; 2020.**

| S.N | Variables | Categories | Frequency | Percentages |
|-----|-----------|-----------|-----------|-------------|
| 1. | Age | 18–29 | 375 | 31.4% |
| | | 30–44 | 452 | 37.9% |
| | | 45–59 | 206 | 17.3% |
| | | 60–69 | 105 | 8.8% |
| | | > = 70 | 53 | 4.5% |
| 2. | Sex | Female | 644 | 54.% |
| | | Male | 547 | 46% |
| 3. | Ethnicity | Oromo | 847 | 71.2% |
| | | Amhara | 116 | 9.7% |
| | | Agnua | 103 | 8.6% |
| | | Nuer | 91 | 7.6% |
| | | Others | 34 | 2.9% |
| 4. | Religion | Orthodox | 433 | 36.4% |
| | | Protestant | 387 | 32.6% |
| | | Muslim | 356 | 29.9% |
| | | Catholic | 13 | 1.1% |

**Table 2. Bi-variable and multivariable logistic regression of factors associated with non-adherence to physical activity recommendations among adults in Southwest Ethiopia, 2020.**

| S.N | Variables | Variable Categories | Adherence to PA recommendations | | COR(95%CI) | AOR(95%CI) | p-value |
|---|---|---|---|---|---|---|---|
| | | | Non-adherent | Adherent | | | |
| 1. | Sex | Male | 240 | 297 | Reference | Reference | |
| | | Female | 489 | 165 | 3.6(2.4–5.6) | 6.1(3.5–10.5) | <0.001* |
| 2. | Age | 18–29 | 219 | 169 | Reference | Reference | |
| | | 30–44 | 246 | 210 | 0.95(.6–1.5) | 1.2(0.66–2.3) | 0.499 |
| | | 45–59 | 141 | 60 | 1.8(.98–3.4) | 2.5(1.05–5.9) | 0.036* |
| | | 60–69 | 69 | 21 | 2.5(1.02–6.4) | 6.6(2.3–19) | 0.001* |
| | | > = 70 | 56 | 9 | 4.6(1.3–16.6) | 6.9(1.5–32.0) | 0.014* |
| 3. | Marital Status | Married | 282 | 174 | Reference | Reference | - |
| | | Others | 447 | 288 | 0.9(.63–1.4) | - | - |
| 4. | Educational Status | Primary and Below | 318 | 180 | 1.2(0.8–1.83) | 0.51(0.3–0.9) | 0.03* |
| | | Secondary and Above | 411 | 282 | Reference | Reference | |
| 5. | Group Membership | Not member of group | 150 | 99 | .95(.57–1.5) | - | - |
| | | Member of a group | 579 | 363 | Reference | - | - |
| 6. | House condition (Crowding) | No crowding | 672 | 441 | Reference | - | - |
| | | Over crowding | 57 | 21 | 1.7(.73–4.3) | - | - |
| 7. | Wealth Index | 1$^{st}$ quartile | 159 | 135 | Reference | Reference | |
| | | 2$^{nd}$ Quartile | 201 | 99 | 1.7(.96–3.0) | 1.7(0.8–3.5) | 0.147 |
| | | 3$^{rd}$ Quartile | 204 | 96 | 1.8(1.01–3.2) | 1.24(0.6–2.6) | 0.566 |
| | | 4$^{th}$ Quartile | 165 | 132 | 1.06(.6–1.8) | 0.7(0.33–1.5) | 0.36 |
| 8. | Level of Happiness | Happy | 645 | 429 | Reference | Reference | |
| | | Neither | 57 | 15 | 2.52(0.9–7.0) | 4.7(1.3–16.8) | 0.016* |
| | | Unhappy | 27 | 18 | 0.99(.34–2.8) | 0.13(0.01–0.9) | 0.049* |
| 9. | Active Family | No | 462 | 198 | 2.3(1.5–3.5) | 2.5(1.4–4.3) | 0.001* |
| | | Yes | 267 | 264 | Reference | Reference | |
| 10. | Active Friends | No | 537 | 213 | 3.2(2.1–5.0) | - | - |
| | | Yes | 192 | 249 | Reference | - | - |
| 11. | Participation in Community activities | No | 207 | 63 | 2.5(1.5–4.3) | 2.7(1.3–5.5) | 0.007* |
| | | Yes | 522 | 399 | Reference | Reference | |

*Significant association;—did not appear in the final step of multivariable logistic regression

CI [3.5–10.5]. In addition, participants who are primary and below in their educational statuses have 50% reduction in probability of being non-adherent to PAR; AOR = 0.51: 95% CI [.28–0.93].

Participants whose none of their family members do physical exercises are nearly three times more likely to be non-adherent to PAR when compared with participants who have family members who do physical exercises; AOR = 2.5: 95% CI [1.4–4.3]. Participating in community-wide activities appears to be protective against physical inactivity. In this study participants who did not participate in community activities are nearly three times non-adherent to PAR when compared with participants who participated in such activities; AOR = 2.7:95%CI [1.3–5.5].

In this study, self-reported feeling of happiness is also associated with being non-adherent to PAR. Participants who reported to be unhappy in their daily life are roughly five times more non-adherent to PAR when compared with happy counterparts AOR = 4.7: CI [1.3–16.8].

## Discussion

In the present study, majority of the study participants (61.2%), 95% CI: (56.2–66.0) are non-adherent to PAR. Older adults and females are more non-adherent to the recommendation when compared with their counterparts.

Level of non-adherence to PAR in the study area is very high when compared with the national prevalence, which reported the prevalence of non-adherence to PAR as 6%. The higher prevalence from the current study may be explained by the difference in the sampled population and sample size. The current study only included urban residents while the national survey included sample of citizens from both urban and rural settings. The population distribution of Ethiopia is predominantly rural with agriculture as a primary source of income. Participation in farm activities demand moderate to vigorous intensity PA which may have lowered the prevalence of PI at the national level. In addition, the relatively large sample size for national survey may also have an impact on the magnitude of non-adherence [18, 22].

The prevalence of no-adherence to PAR in the study area is also higher than reports from Papua New Guinea where 34% of participants are not-adherent to PAR. The current finding is also higher than a report from India among adults in Kani tribe where only 9.7% are non-adherent. The relatively lower prevalence of non-adherence to the recommendation in Papua New Guinea and Kani tribe of India may be due to their lifestyle that involve rural agriculture, which tends to increase the time for physical activity [10, 11].

In contrary, the prevalence of non-adherence to PAR from the current study is lower than reports from Nigeria that showed the prevalence of non-adherence to PAR as 77.8% among students and 69.7% among civil servants. The relatively higher prevalence in Nigeria may be due to the type of participants included in the studies. The current study involved urban residents while the studies in Nigeria were conducted on civil servants and university students [12, 13]. The finding of the current study is also lower than results of STEPS survey in Kenya where more than 80% are not adhering to the recommendation. The relatively higher level of non-adherence to PAR in Kenya may be due to its relative industrialization and digitalization when compared with Ethiopia [14].

Several factors were associated with non-adherence to PAR. These include older age, female sex, and educational status, family history of physical activity; community participation and self-reported level of happiness.

This study found that older adults are nearly seven times more likely to be non-adherent to PAR when compared with younger adults. The relative disadvantage of older adults in the physical activity is well documented in several studies. This may be due to unsuitability of their environment for regular exercise and fear of falling while doing physical exercise [23–25].

In this study, females are six times more likely to be non-adherent to PAR when compared with male participants. Several studies also reported similar findings indicating more likelihood of non-adherence to PAR among females [26–28] This may be due to less involvement of urban women and housewives on works that require moderate to vigorous intensity exercise unlike rural women who are involved in farm activities. Urban women tend to be at home or office, working less vigorous works [29]. Employment pattern of women may also pose females at risk of staying home, which in turn reduce the time used to perform physical activities. According to the Central Statistics Agency of Ethiopia, only 41% of women who are expected to work are employed [22].

This study revealed that participants with lower educational achievement have reduced probability of non-adherence to PAR that reaches up to 50%. This finding contradicts with a report from Malaysia that reported up to 20% increase in physical inactivity among participants with lower educational achievement [9]. This may be due to engagement of peoples with

lower educational status in informal employment activities. Informal employment activities tend to require more moderate to vigorous intensity physical activities when compared to office based jobs. A recent study in Australia indicated that desk based working environments are not adequately equipped with facilities that encourage movement for office workers [30].

Participants without physically active family member are more likely to be non-adherent to PAR when compared with participants who have physically active family members. Families are essential components of one's assets that cue peoples to practice healthy behaviors and refrain from something that may harm health. Studies in different parts of the world also acknowledged the importance of families on adopting a regular PA plan among school aged children [31–34]. Frequent encouragement and support from family members has also resulted in reduced leisure time sedentariness [35].

Participants who were involved in community based communal activities have lower probability of being non-adherent to PAR. Similarly, participants who do not participate in such community based activities tend to be more physically inactive. This may be due to the overall effect of social participation on health and wellbeing [36]. Community participation is found to enhance health through several means; among these is enhancement of mental health that in turn causes to take preventive measures against chronic diseases. PA is one of the essential activities to prevent the occurrence of chronic diseases [37].

Participants who self-reported to be unhappy in their current conditions are also more likely to be non-adherent to PAR when compared with happier ones. This unfavorable mental status may result in reduced intention to take proactive measures in maintaining health by performing PA. Such status is also an indicator of poor social capital, and social capital is found to be extremely important in physical activities as shown from many studies [38, 39].

Generally, this study found a high prevalence of non-adherence to PAR among urban residents in Southwest Ethiopia. This could result in consequent increase in the incidence and prevalence of NCDs in the near future, if the current trend continues. Old-aged urban residents, females, relatively educated residents are more likely to be non-adherent to the recommendation. In addition, not participating in community-wide activities and low level of perceived happiness is also associated with non-adherence to PAR.

## Limitations

This study assessed the level of adherence to PAR depending on respondent's response to set of questions using the GPAQ. The tool fails to objectively measure the level of adherence unlike other advanced tools. In addition, the nature of questionnaire may have introduced recall bias as subjects may forget some of their physical movements.

## Conclusion

Majority of the study participants failed to meet the global recommendations for PA. This high level of physical inactivity poses the risk of future occurrence of NCDs and other health problems. Age, sex, educational status, community engagement, level of happiness and PA of family members were associated with non-adherence to PAR.

Community based interventions to raise the level of PA and boost adherence to the recommendation has to be implemented to protect the health of the community from the looming danger posed by PI. Priority has to be given to old aged adults and females. Community organizations and networks have to be used to encourage PA among members. Family based changes have to be fostered to reduce PI by using family members as role models. Strengthening social capital and social cohesion also helps to tackle the problem in the long run.

## Supporting information

**S1 Data. Data supporting the findings.**
(SAV)

## Acknowledgments

We thank all who helped us in realization of this research work. We thank Mettu University for providing material and financial assistance to this study. In addition we extend our gratitude to data collectors, supervisors and study participants for their cooperation throughout the conduct of this work.

## Author Contributions

**Conceptualization:** Sabit Zenu, Endegena Abebe, Mohammed Reshad, Yohannes Dessie, Rukiya Debalke, Tsegaye Berkessa.

**Data curation:** Sabit Zenu, Endegena Abebe, Mohammed Reshad, Yohannes Dessie, Rukiya Debalke, Tsegaye Berkessa.

**Formal analysis:** Sabit Zenu, Endegena Abebe, Mohammed Reshad, Yohannes Dessie, Rukiya Debalke, Tsegaye Berkessa.

**Funding acquisition:** Sabit Zenu, Endegena Abebe, Mohammed Reshad, Yohannes Dessie, Rukiya Debalke, Tsegaye Berkessa.

**Investigation:** Sabit Zenu, Endegena Abebe, Mohammed Reshad, Yohannes Dessie, Rukiya Debalke, Tsegaye Berkessa.

**Methodology:** Sabit Zenu, Endegena Abebe, Mohammed Reshad, Yohannes Dessie, Rukiya Debalke, Tsegaye Berkessa.

**Project administration:** Sabit Zenu, Endegena Abebe, Mohammed Reshad, Yohannes Dessie, Rukiya Debalke, Tsegaye Berkessa.

**Resources:** Sabit Zenu, Endegena Abebe, Mohammed Reshad, Yohannes Dessie, Rukiya Debalke, Tsegaye Berkessa.

**Software:** Sabit Zenu, Endegena Abebe, Mohammed Reshad, Yohannes Dessie, Rukiya Debalke, Tsegaye Berkessa.

**Supervision:** Sabit Zenu, Endegena Abebe, Mohammed Reshad, Yohannes Dessie, Rukiya Debalke, Tsegaye Berkessa.

**Validation:** Sabit Zenu, Endegena Abebe, Mohammed Reshad, Yohannes Dessie, Rukiya Debalke, Tsegaye Berkessa.

**Visualization:** Sabit Zenu, Endegena Abebe, Mohammed Reshad, Yohannes Dessie, Rukiya Debalke, Tsegaye Berkessa.

**Writing – original draft:** Sabit Zenu, Endegena Abebe, Mohammed Reshad, Yohannes Dessie, Rukiya Debalke, Tsegaye Berkessa.

**Writing – review & editing:** Sabit Zenu, Mohammed Reshad, Yohannes Dessie, Rukiya Debalke, Tsegaye Berkessa.

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
