## [Decision Letter · Decision Letter 0]

6 May 2022

PGPH-D-22-00157

Non-adherence to the World Health Organization’s physical activity recommendations and associated factors among healthy adults in urban centers of Southwest Ethiopia

Dear Dr. Zenu,

Thank you for submitting your manuscript to PLOS Global Public Health. After careful consideration, we feel that it has merit but does not fully meet PLOS Global Public Health’s publication criteria as it currently stands. Therefore, we invite you to submit a revised version of the manuscript that addresses the points raised during the review process.

Please submit your revised manuscript by . If you will need more time than this to complete your revisions, please reply to this message or contact the journal office at globalpubhealth@plos.org. Please include the following items when submitting your revised manuscript:

We look forward to receiving your revised manuscript.

Kind regards,

Chaisiri Angkurawaranon

Academic Editor

Journal Requirements:

1. Please insert an Ethics Statement at the beginning of your Methods section, under a subheading 'Ethics Statement'. It must include:

- (for human participants/donors) - A statement that formal consent was obtained (must state whether verbal/written) OR the reason consent was not obtained (e.g. anonymity). NOTE: If child participants, the statement must declare that formal consent was obtained from the parent/guardian.

- State the initials, alongside each funding source, of each author to receive each grant.

3. If you have no competing interests to declare, please state "The authors have declared that no competing interests exist" in the Competing Interest Section.

4. In the online submission form, you indicated "All data for this research article is available and can be accessed from the corresponding author.". All PLOS journals now require all data underlying the findings described in their manuscript to be freely available to other researchers, either 1. In a public repository, 2. Within the manuscript itself, or 3. Uploaded as supplementary information.

Additional Editor Comments (if provided):

There are major concerns raised by the reviewers. Please address them carefully as this will require another round of review

Reviewers' comments:

Reviewer's Responses to Questions

**Comments to the Author**

1. Does this manuscript meet PLOS Global Public Health’s publication criteria? Is the manuscript technically sound, and do the data support the conclusions? The manuscript must describe methodologically and ethically rigorous research with conclusions that are appropriately drawn based on the data presented.

Reviewer #1: Yes

Reviewer #2: No

Reviewer #3: Partly

2. Has the statistical analysis been performed appropriately and rigorously?

Reviewer #1: Yes

Reviewer #2: No

Reviewer #3: Yes

3. Have the authors made all data underlying the findings in their manuscript fully available (please refer to the Data Availability Statement at the start of the manuscript PDF file)?

Reviewer #1: Yes

Reviewer #2: No

Reviewer #3: Yes

4. Is the manuscript presented in an intelligible fashion and written in standard English?

Reviewer #1: Yes

Reviewer #2: Yes

Reviewer #3: Yes

5. Review Comments to the Author

Reviewer #1: The manuscript titled “Non-adherence to the World Health Organization’s physical activity recommendations and associated factors among healthy adults in urban centers of Southwest Ethiopia” deals with an important issue of health and well-being. The purpose of this study is to To determine the prevalence of non-adherence to physical activity recommendations and identify its associated factors among healthy adults in urban centers of Southwest Ethiopia.

This is an interesting work. It is my pleasure to give my insights and highlight some recommendations and commentaries to the manuscript, considering my expertise in the field. I really appreciate this opportunity to discuss the evidence and I hope my contributions could serve to improve the final version of the study.

Please highlight better in all text sections (abstract, introduction, conclusion) the scientific/clinical relevance of your work. Please provide a clear “take-home message” of the importance of this paper in the scientific community and the novelty of this paper according to the current literature, to help better readers understanding in the conclusion section.

Pag 3, line 62. Please add in all kind of individuals and quote adequate references as follow: Physical Activity for Health—An Overview and an Update of the Physical Activity Guidelines of the Italian Ministry of Health. J. Funct. Morphol. Kinesiol. 2016, 1, 269-275. https://doi.org/10.3390/jfmk1030269

Reviewer #2: Thanks for the opportunity to read this article.

When I saw the article, I had great expectations because it was done with a population that has been little studied. However, after reading it, I have several comments that I present below.

Line 29. What type of factors did you try to identify? This information has to be clear.

Line 31. From where?

Line 35. “In this study” can be deleted.

Line 35. “are”, the verbs must be in the past. Revise the entire document.

Lines 49-53. The entire document must be revised. The sentences are about the present, but the references to support the statement are quite old. The sentences and the references must be accurate.

Line 54. The first sentence needs a reference.

Lines 58-62. These sentences need the references of the WHO. Moreover, the recommendation is not accurately described. The text must be amended.

Revise the entire introduction. References have to be put after the sentences, not just at the end of the paragraphs.

There is no information about the factors related to PA in the methods section. This information is important. The readers have to know the variables and how the data were collected. This is a big flaw.

Lines 139-141. How ethnicity and religion were asked?

Lines 147-164. It is not clear how these results appear in the article. There is no reference to these factors in the methods, and it is not clear which instruments were used to measure them.

I didn't read the discussion carefully because this article has flaws that I consider serious. I highlight two. 1. The methods are not exhaustive in explaining all the variables that enter the analyses. 2. The theoretical rationale is non-existent. For example, nothing is said about which factors will be studied (e.g. sociodemographic, social, psychological factors, environments, among others).

Reviewer #3: Thank you for inviting me to review the manuscript entitled ‘Non-adherence to the World Health Organization’s physical activity recommendations and associated factors among healthy adults in urban centers of Southwest Ethiopia’. The manuscript presents the prevalence of physical inactivity in urban residents in Ethiopia. Please consider my comments.

Abstract:

- Please consider specifying the details of educational status and level of happiness to present the AOR. For example, female sex had the higher odds to be physically inactive. How about educational status and level of happiness (lower is a risk or a preventive factor)?

Introduction:

- This section consists of 8 paragraphs. In my opinion, they are too many. Some short paragraphs present the similar point which can be combined into one paragraph.

- In this section, there are three important terms, ‘physical activity’, ‘physical inactivity’, and ‘non-adherence to physical activity recommendations’. In this context, physical inactivity = non-adherence to physical activity recommendations. I would recommend the authors to define these terms, then, use them in a consistent manner through the entire manuscript.

- Please consider citing references to specific sentences instead of citing a whole paragraph. For example, paragraph 3 (line 58-69), the reference no. 2-5 should be separated and cited after the relevant sentences. Please check through the entire manuscript.

Methods:

- I would recommend the authors to break down this section into sub-sections. A reporting guideline (e.g., STORBE: https://www.equator-network.org/reporting-guidelines/strobe/) may be helpful to organise the contents.

- Please consider presenting the details of the questionnaire. It seems the questionnaire included other contents beyond the GPAQ. I strongly recommend presenting the validity and reliability of the data collection tool.

- Please present the major variables collected by the questionnaire and how to analyse the variables. The methods for analysing the data collected by GPAQ should be described.

Results:

- Please consider combining Table 2 and Table 3. The contents in Table 2 are quite similar to Table 3. The percentages of non-adherent and adherent should be inserted for all variables in Table 3.

- Table 3: why are some AORs missing? For example, the COR of educational status was 1.2 (0.8-1.83) (non-significant), its AOR was calculated. However, the AOR of marital status was not present.

Discussion:

- Line 178-183: Why did the authors choose Papua New Guinea and India as comparators? I would recommend the authors to emphasise the points of comparisons, for example, comparing between the study findings with other studies in Ethiopia and in the African region. Otherwise, comparing between urban and rural populations in other countries may be helpful to explain the differences of this current study and the previous study conducted in rural Ethiopia.

- Please consider the limitations of this study. The present limitation is unclear. What is the main problem of GPAQ compared with an objective measurement of physical activity? Are there any other limitations?

References:

- Please check the consistency of the references (e.g., journal abbreviations, small and capital letters).

General comments:

- Please recheck the writing style/typos (e.g., don’t, didn’t, peoples, missing full stop (line 202)).

6. PLOS authors have the option to publish the peer review history of their article (what does this mean?). If published, this will include your full peer review and any attached files.

**Do you want your identity to be public for this peer review?** For information about this choice, including consent withdrawal, please see our Privacy Policy.

Reviewer #1: No

Reviewer #2: No

Reviewer #3: No

---

## [Decision Letter · Decision Letter 1]

27 Sep 2022

PGPH-D-22-00157R1

Non-adherence to the World Health Organization’s physical activity recommendations and associated factors among healthy adults in urban centers of Southwest Ethiopia

Dear Dr. Zenu,

Thank you for submitting your manuscript to PLOS Global Public Health. After careful consideration, we feel that it has merit but does not fully meet PLOS Global Public Health’s publication criteria as it currently stands. Therefore, we invite you to submit a revised version of the manuscript that addresses the points raised during the review process.

We look forward to receiving your revised manuscript.

Kind regards,

Chaisiri Angkurawaranon

Academic Editor

Journal Requirements:

Additional Editor Comments (if provided):

Reviewers' comments:

Reviewer's Responses to Questions

**Comments to the Author**

1. If the authors have adequately addressed your comments raised in a previous round of review and you feel that this manuscript is now acceptable for publication, you may indicate that here to bypass the “Comments to the Author” section, enter your conflict of interest statement in the “Confidential to Editor” section, and submit your "Accept" recommendation.

Reviewer #3: (No Response)

2. Does this manuscript meet PLOS Global Public Health’s publication criteria? Is the manuscript technically sound, and do the data support the conclusions? The manuscript must describe methodologically and ethically rigorous research with conclusions that are appropriately drawn based on the data presented.

Reviewer #3: Yes

3. Has the statistical analysis been performed appropriately and rigorously?

Reviewer #3: Yes

4. Have the authors made all data underlying the findings in their manuscript fully available (please refer to the Data Availability Statement at the start of the manuscript PDF file)?

Reviewer #3: Yes

5. Is the manuscript presented in an intelligible fashion and written in standard English?

Reviewer #3: Yes

6. Review Comments to the Author

Reviewer #3: I wish to thank you the authors for your efforts to revise the manuscript. In terms of the clarity of the manuscript, this version has been improved. However, I still have some suggestions with regard to the previous comments for consideration.

Introduction:

- Please consider reorganising this section. The current introduction consists of 8 paragraphs. Some paragraphs are very brief. Is it possible to combine some paragraphs together and make it more concise?

Results:

- Sub-section ‘Non-adherence to the WHO recommendations of physical activity’ – The old Table 2 presenting a chi-square test has been removed. Therefore, I would like to recommend the authors to add brief explanation of the chi-square test in the text.

- Page 7, line 165: please replace Table 3 with Table 2 (the old Table 3 has been removed already).

- Previously, I enquired about missing AORs. The authors explained that ‘only the variables with significant association in the final model were presented with the AOR values’. . I am confused by reading the data in the current Table 2. For example, Marital status (other): COR = 0.9(.63-1.4) – no AOR, while Educational Status (Primary and Below): COR 1.2(0.8-1.83), 0.51(0.3-0.9), p = 0.03* [is this COR significant?]. The wealth index (2nd to 4th) quartiles were not significant, however, their AORs were calculated. Please clarify and organise the numbers, if appropriate.

References:

- Please check the consistency of the references (e.g., journal abbreviations, small and capital letters). For example, ref#2: Global Recommendations on Physical Activity for Health for age 18- 64 years old (using capital letters at the beginning of each word) VS. ref#3: Global action plan on physical activity 2018-2030: more active people for a healthier world (using small letters except the first word). This should be consistent throughout the reference list.

General comments:

- Please recheck the writing style/typos (e.g., don’t, didn’t).

7. PLOS authors have the option to publish the peer review history of their article (what does this mean?). If published, this will include your full peer review and any attached files.

**Do you want your identity to be public for this peer review?** For information about this choice, including consent withdrawal, please see our Privacy Policy.

Reviewer #3: No

---

## [Decision Letter · Decision Letter 2]

13 Dec 2022

Non-adherence to the World Health Organization’s physical activity recommendations and associated factors among healthy adults in urban centers of Southwest Ethiopia

PGPH-D-22-00157R2

Dear Mr. Zenu,

We are pleased to inform you that your manuscript 'Non-adherence to the World Health Organization’s physical activity recommendations and associated factors among healthy adults in urban centers of Southwest Ethiopia' has been provisionally accepted for publication in PLOS Global Public Health.

Best regards,

Chaisiri Angkurawaranon

Academic Editor

Reviewer Comments (if any, and for reference):

Reviewer's Responses to Questions

**Comments to the Author**

1. If the authors have adequately addressed your comments raised in a previous round of review and you feel that this manuscript is now acceptable for publication, you may indicate that here to bypass the “Comments to the Author” section, enter your conflict of interest statement in the “Confidential to Editor” section, and submit your "Accept" recommendation.

Reviewer #3: All comments have been addressed

2. Does this manuscript meet PLOS Global Public Health’s publication criteria? Is the manuscript technically sound, and do the data support the conclusions? The manuscript must describe methodologically and ethically rigorous research with conclusions that are appropriately drawn based on the data presented.

Reviewer #3: Yes

3. Has the statistical analysis been performed appropriately and rigorously?

Reviewer #3: Yes

4. Have the authors made all data underlying the findings in their manuscript fully available (please refer to the Data Availability Statement at the start of the manuscript PDF file)?

Reviewer #3: Yes

5. Is the manuscript presented in an intelligible fashion and written in standard English?

Reviewer #3: Yes

6. Review Comments to the Author

Reviewer #3: I wish to thank the authors for revising the manuscript according to my previous comments. I would recommend this manuscript for publication.

7. PLOS authors have the option to publish the peer review history of their article (what does this mean?). If published, this will include your full peer review and any attached files.

**Do you want your identity to be public for this peer review?** For information about this choice, including consent withdrawal, please see our Privacy Policy.

Reviewer #3: No
